# Phytomedicines to Target Hepatitis B Virus DNA Replication: Current Limitations and Future Approaches

**DOI:** 10.3390/ijms23031617

**Published:** 2022-01-30

**Authors:** Rahila Zannat Sadiea, Shahnaj Sultana, Bijan Mohon Chaki, Tasnim Islam, Sharmy Dash, Sharmin Akter, Md Sayeedul Islam, Taheruzzaman Kazi, Abir Nagata, Rocco Spagnuolo, Rosellina Margherita Mancina, Md Golzar Hossain

**Affiliations:** 1Department of Microbiology and Hygiene, Bangladesh Agricultural University, Mymensingh 2202, Bangladesh; rsadiea@gmail.com (R.Z.S.); triptishahnaj@gmail.com (S.S.); btasnimislam@gmail.com (T.I.); 2Department of Chemistry (Organic Chemistry Division), Begum Rokeya University, Rangpur 5400, Bangladesh; drbijan.chem@brur.ac.bd; 3Department of Pharmacology, Bangladesh Agricultural University, Mymensingh 2202, Bangladesh; shormydas@gmail.com; 4Department of Physiology, Bangladesh Agricultural University, Mymensingh 2202, Bangladesh; sharmin.akter@bau.edu.bd; 5Department of Biological Sciences, Graduate School of Science, Osaka University, Osaka 560-0043, Japan; islam@bio.sci.osaka-u.ac.jp; 6Department of Regenerative Dermatology, Graduate School of Medicine, Osaka University, Suita 565-0871, Japan; tanjim@r-derma.med.osaka-u.ac.jp (T.K.); abir85n@r-derma.med.osaka-u.ac.jp (A.N.); 7Experimental and Clinical Medicine Department, Magna Graecia University, 88100 Catanzaro, Italy; spagnuolo@unicz.it; 8Department of Molecular and Clinical Medicine, Göteborgs Universitet, 413 45 Gothenburg, Sweden; rosellina.mancina@wlab.gu.se

**Keywords:** hepatitis B virus, DNA replication, phytomedicines/phytochemicals, limitations, future approaches

## Abstract

Hepatitis B virus infection (HBV) is one of the most common causes of hepatitis, and may lead to cirrhosis or hepatocellular carcinoma. According to the World Health Organization (WHO), approximately 296 million people worldwide are carriers of the hepatitis B virus. Various nucleos(t)ide analogs, which specifically suppress viral replication, are the main treatment agents for HBV infection. However, the development of drug-resistant HBV strains due to viral genomic mutations in genes encoding the polymerase protein is a major obstacle to HBV treatment. In addition, adverse effects can occur in patients treated with nucleos(t)ide analogs. Thus, alternative anti-HBV drugs of plant origin are being investigated as they exhibit excellent safety profiles and have few or no side effects. In this study, phytomedicines/phytochemicals exerting significant inhibitory effects on HBV by interfering with its replication were reviewed based on different compound groups. In addition, the chemical structures of these compounds were developed. This will facilitate their commercial synthesis and further investigation of the molecular mechanisms underlying their effects. The limitations of compounds previously screened for their anti-HBV effect, as well as future approaches to anti-HBV research, have also been discussed.

## 1. Introduction

Hepatitis B infection, which is caused by the hepatitis B virus (HBV), is a common life-threatening, infectious disease that affects the liver. When the disease evolves from an acute infection to chronic hepatitis, it can result in cirrhosis, hepatocellular carcinoma (HCC), and eventually death if not properly treated. Although vaccination is an effective means of preventing HBV infection, according to the World Health Organization’s (WHO) 2021 report, 296 million individuals are still living with chronic hepatitis B worldwide [1]. In African and Western Pacific countries, 6.1% and 6.2% of the adult population, respectively, are infected with HBV [2]. In contrast, the prevalence of HBV infection in the European, South-East Asian, and Eastern Mediterranean countries is 1.6%, 2.0%, and 3.3%, respectively [3]. Therefore, the WHO has implemented an action plan to eradicate HBV by 2030, and this involves the identification of novel and alternative drugs against the disease.

HBV is a small enveloped DNA virus of the Hepadnaviridae family. It is composed of an outer envelope and an inner core. The outer envelope of the virus contains a surface protein called the surface antigen or HBsAg. The 3.2 kb, partially double-stranded HBV genome DNA (dsDNA) encodes four overlapping open reading frames (ORFs), i.e., P, S, C, and X [4]. The viral polymerase, which is encoded by ORF P, contains the reverse transcriptase (RT) domain, which plays a vital role in viral replication and pathogenesis [5,6]. Nucleos(t)ide analogs such as lamivudine, adefovir dipivoxil, telbivudine, entecavir, tenofovir, and clevudine can suppress HBV replication by targeting HBV DNA polymerase [7,8]. However, these drugs induce serious adverse effects in patients; moreover, the virus may develop resistance to these nucleos(t)ide inhibitors due to the occurrence of spontaneous mutations [9]. In addition, interferons (INF-α and PegIFN-α) are used for the treatment of the HBV infection [10] as they are believed to affect viral replication and increase cell-mediated immunity in the host. However, their application is limited due to the low curing rates and the occurrence of serious side effects [10]. However, a recent study showed that HBV replication is suppressed by the interleukin 1 beta (IL-1β) produced from the macrophages activated by the HBV [11]. The IL-1β-mediated suppression of HBV replication has further been proven both in Huh7 cells and HBV transgenic mice [11]. This IL-1β treatment might be another option to treat and control the HBV infection.

Vaccination is effective in preventing HBV infection in most cases but is ineffective in individuals who have contracted the virus [12]. A recent study evaluated the persistence of the long-term immunogenicity of HBV vaccine, in order to establish the main associated factors and determinants of the disease. The results showed that the HBV vaccine is effective even 20 years after vaccination [13]. Although vaccination is routinely practiced in most countries, new infections are still reported in several countries, including Bangladesh, China, Georgia, Haiti, India, Kenya, and Mauritania, among other countries [1,14,15]. These new HBV infections occur either due to non-vaccination or vaccination failure due to viral genomic mutations [16,17]. In addition, drug-resistant HBV strains, which are more virulent than wild-type strains, have been circulating, leading to an increase in HCC incidence and related death rates [16,18,19]. In addition, chronic HBV patients may transmit the disease to healthy individuals either vertically or horizontally through unprotected sexual intercourse, sharing of syringes and needles, or blood donation, among other routes [20]. However, recent investigation focuses on vertical and horizontal transmission to provide the epidemiological profile of HBV and evaluating the long-term effects of the available vaccines [21]. Therefore, scientists are constantly seeking alternative medicines against drug-resistant HBV. Over the years, various phytomedicines and phytochemicals have been screened for potential anti-HBV effects in various cell culture systems (in vitro) and in transonic mice or ducklings (in vivo), because of their better safety profile, with few or no side effects [19,22]. Although phytomedicines and phytochemicals can inhibit HBsAg proliferation and secretion, the inhibition of DNA replication is the ideal mechanism for the effective curing of the disease. However, very few reviews have summarized the phytomedicines and phytochemicals with significant effects against HBV. Therefore, this review primarily focused on important phytomedicines and phytochemicals that exhibit significant inhibitory effects against HBV by interfering with its DNA replication. In addition, the chemical structures of the various compounds of plant origin have been generated and presented.

We used the PubMed, PubMed Central, PubChem, and Scopus search engines, among others, to search for published research articles that investigated compounds of plant origin and their effects against HBV DNA replication, and we reviewed them thoroughly. ChemDraw Professional 16.0 was used to draw all chemical structures.

## 2. Phytomedicines and Phytochemicals Regulating HBV DNA Replication

### 2.1. Terpenoid Compounds

Terpenes are linear or cyclic compounds consisting of five-carbon isoprene basic structural units (saturated or unsaturated). Isoprene units can assemble in several different ways, thereby resulting in a wide variety of secondary metabolites. Terpenoids are modified terpenes with different functional groups, and oxidized methyl groups are moved or removed at various positions in these compounds. With varying structures that result in diverse functionality, terpenoids are potential biologically active compounds, and many of them exhibit inhibitory activities against various human cell lines; for example, Taxol and its derivatives are used as anticancer drugs due to their inhibitory effects on cancer cell proliferation [23]. The structures of various terpenoid compounds that interfere with HBV DNA replication are presented in Figure 1. A triterpenoid saponin (**1**) was extracted from the Tibetan herb, *Potentilla anserina*, using ethanol, and its in vivo anti-HBV effects were evaluated in Peking ducklings. Reportedly, this compound inhibited duck hepatitis B (DHBV) DNA replication [24]. Furthermore, saponins were extracted from the traditional Chinese herbal medicinal plant, *Abrus cantoniensis* Hance (AC), and their anti-HBV effects were evaluated both in vitro and in vivo [25]. The saponin extract decreased HBV DNA production in both HepG2.2.15 cells and C57BL/6 mice infected with recombinant HBV [25]. In mice treated with the saponins, the percentage of CD4+ and T cells in their spleens and serum IFN-γ levels were also upregulated [25]. Ganoderic acid (**2**), isolated from *Ganoderma lucidum*, inhibited HBV replication in HepG2.2.15 cells when they were treated at 8 µg/mL for up to eight days [26]. Two terpenoid compounds, astataricusone B (**3**) and epishionol (**4**), were isolated from the roots and rhizomes of *Aster tataricus*, and found to exhibit inhibitory activity against HBV DNA replication, with IC_50_ (half-maximal inhibitory concentration) values of 2.7 and 30.7 μM/L, respectively [27]. Hemslecin A (**5**), extracted from flowering plants of the genus *Hemsleya*, inhibited HBV DNA replication, with an IC_50_ value of 11.2 μM (SI = 5.8) in HepG2.2.15 cells [28]. The climbing vine swallowwort, *Cynanchum auriculatum*, reportedly contains the terpenoid compound caudatin (**6**), which inhibits HBV DNA replication, with an IC_50_ value of 40.62 mM/L (SI = 6.0) [29]. Glycyrrhizin (**7**) and its metabolite, glycyrrhetinic acid (GA) (**8**), are the main constituents of *Licorice* roots (*Glycyrrhizae glabra*), and exhibit inhibitory activity against HBV DNA replication (IC_50_ = 39.28 μM/L) [30]. Helioxanthin (HE-145) (**9**) was extracted from the heartwood, *Taiwania cryptomerioides* hayata, and inhibited HBV replication in HCC cells. HE-145 reportedly decreased HBV DNA-binding activity in HepA2 cells, with a unique anti-HBV mechanism, and, thus, is a potential anti-HBV agent [31]. Asiaticoside (**10**), isolated from *Hydrocotyle sibthorpioides*, significantly reduced HBV transcription, replication, as well as cccDNA levels, by suppressing the promoter activities of the core S1, S2, and X genes in HepG2.2.15 cells [32]. In addition, the phytocompound methyl helicterate (MH) (**11**), isolated from the Chinese herb *Helicteres angustifolia*, significantly reduced cccDNA and viral RNA levels in HepG2.2.15 cells and DHBV-infected ducklings [33]. Lupane-type triterpenoids, including betulinic acid (**12**), were isolated from *T. conophorum* seeds, characterized, and their anti-HBV effects were investigated in rat models and HepG2 cells. These compounds showed hepatoprotective and cytotoxic activities, which may be due to their interaction with HBV, as the compounds exhibited high binding affinities for the virus [34].

### 2.2. Flavonoid Compounds

Flavonoids constitute a diverse range of polyphenolic structures found in many natural substances, such as fruits, vegetables, grains, bark, roots, stems, flowers, tea, and wine. Currently, flavonoids are considered indispensable tools in the field of human physiology due to their anti-oxidative, anti-inflammatory, anti-mutagenic, and anti-carcinogenic properties. The flavonoids of medicinal interest can be classified into several groups, such as flavonols, flavones, flavanones, isoflavones, and anthocyanidins [35]. The chemical structures of some flavonoid compounds affecting HBV DNA replication are shown in Figure 2. Camellia sinensis (green tea), which contains epicatechin (ECG), epigallocatechin (EGC), epicatechin (EC), catechin (C), and epigallocatechin gallate (EGCG) (**13**–**17**), exhibits significant antiviral activity through various mechanisms. The antiviral effects of green tea extract (GTE) were evaluated for HBV in Hep2-N10 cells by measuring viral antigen levels and quantifying viral DNA. The extract inhibited extracellular HBV DNA production. Moreover, GTE exhibited more significant antiviral effects than EGCG in HepG2.117 cells. EGCG targets only the replicative intermediates formed during DNA synthesis; for example, it was found to reduce HBV cccDNA production [36,37]. Isovitexin (**18**), a flavonoid isolated from *Swertia yunnanensis*, exhibits significant anti-HBV effects and suppresses HBV DNA replication, with IC_50_ values of 0.09 mM, <0.01 mM, and 0.05 mM [38]. The flavonoid compound LPRP-Et-97543 (**19**), isolated from Liriope muscari (Decne.) L.H.Bailey, exhibited significant anti-HBV activity and significantly suppressed the activities of the core, S, and preS promoters and viral DNA replication by regulating viral proteins [39]. Isooriention (**20**), isolated from Swertia mussotii, inhibited HBV DNA replication, with an IC_50_ value of 0.02 mM [40]. Robustaflavone (**21**), extracted from *Rhus succedanea*, showed potent inhibitory effects against HBV replication in HepG2.2.15 cells, with a 50% effective concentration (EC_50_) of 0.25 mM, and an SI of 153 [41]. Robustaflavone hexaacetate (**22**) inhibited HBV replication, with an EC_50_ value of 0.73 mM/L, whereas wogonin (**23**), isolated from *Scutellaria baicalensis* reduced the levels of relaxed circular and linear forms of HBV DNA [42].

### 2.3. Phenolic and Polyphenolic Compounds

Phenolic and polyphenolic compounds constitute one of the largest and most widespread plant secondary metabolite groups, and they possess significant antiviral and antioxidant properties. Although polyphenols include a wide range of molecules that exhibit different biological activities, their core structure is composed of a benzene ring and a hydroxyl functional group [43]. The chemical structures of various phenolic and polyphenolic compounds of plant origin, which affect HBV DNA replication, are presented in Figure 3. The in vitro anti-HBV effects of the polyphenolic extract of *Geranium carolinianum* L. (PPGC) were evaluated in HepG2.2.15 cells, and the extract was found to decrease HBsAg and HBeAg secretion, with IC_50_ values of 46.85 μg/mL and 65.60 μg/mL, respectively. Southern blot analysis further confirmed that PPGC decreased plasma and liver DHBV DNA levels in infected ducklings in a dose-dependent manner [44]. The plant *Oenanthe javanica* (OJ) is widely used for the treatment of hepatitis; the HepG2.2.15 cell line and a DHBV infection model were used as in vitro and in vivo models, respectively, to investigate the anti-HBV effects of its total phenolic extract. It was found that OJ significantly inhibited HBV replication in HepG2.2.15 cells and inhibited DHBV replication in ducks [45]. Several quinic acid derivatives, including 3,4-*O*-dicaffeoylquinic acid, 3,5-*O*-dicaffeoylquinic acid, 3,5-*O*-dicaffeoyl-muco-quinic acid, 5-*O*-caffeoylquinic acid, 3-*O*-caffeoylquinic acid, and 5-*O*-(*E*)-*p*-coumaroylquinic acid (**24**–**29**), extracted from the aerial parts of *Lactuca indica* L. (Compositae), effectively decreased HBV DNA levels in HepG2.2.15 cells [46]. Chlorogenic acid (**30**) and its related compounds, which are found in the leaves and fruits of dicotyledonous plants such as the coffee plant, exhibit antiviral activity. Chlorogenic acid, quinic acid (**31**), and caffeic acid (**32**) reportedly inhibit HBV DNA replication in the HepG2.2.15 cell line [47]. Protocatechuic aldehyde (PA) (**33**), isolated from *Salvia miltiorrhiza*, reduced HBV DNA release in HepG2.2.15 cells in a dose- and time-dependent manner. At the doses of 25, 50, or 100 mg/kg (administered intraperitoneally, twice daily), PA reduced viremia in ducks infected with DHBV [48]. Mulberrofuran G (**34**), isolated from the root bark of *Morus alba* L., exhibited moderate inhibitory activity against HBV in HepG2.2.15 cells by inhibiting its DNA replication, with an IC_50_ value of 3.99 μM/L [49]. Similarly, *p*-Hydroxyacetophenone (p-HAP) (**35**), isolated from *Artemisia morrisonensis*, exhibited inhibitory activity against HBV DNA replication, with an IC_50_ value of 306.4 μM/L [50].

### 2.4. Enyne Compounds

Enynes, which are organic compounds consisting of a carbon–carbon double bond (C=C) and a carbon–carbon triple bond, can inhibit HBsAg and HBeAg secretion. The chemical structures of some enyne compounds affecting HBV DNA replication are shown in Figure 4. *Artemisia capillaris* (Yin-Chen) contains enynes (**36**–**37**), and its extracts were evaluated in HepG2.2.15 cells to determine the active components responsible for its anti-HBV effects. The results showed that the compound atractylodin (**38**), isolated from this herb, significantly inhibited HBV DNA replication, with an IC_50_ value of 9.8 μM (SI > 102), and the hydroxyl and glycosyl groups were found to be responsible for maintaining this activity [51].

### 2.5. Lactone Compounds

The intramolecular condensation of hydroxycarboxylic acids leads to the formation of cyclic esters known as lactones. Most lactones that exhibit biological activity are macrocyclic [52]. Dehydroandrographolide (**39**) (Figure 5) and andrographolide (**40**), isolated from *Andrographis paniculata*, reportedly exhibited inhibitory activity against HBV DNA replication in HepG2.2.15 cells, with IC_50_ values of 22.58 and 54.07 μM/L and low SI values of 8.7 and 3.7 μM/L, respectively [53]. Another lactone compound, artemisinin (**41**), isolated from *Artemisia annua*, inhibited HBV DNA replication in HepG2.2.15 cells, with an IC_50_ value >100 mM/L [54]. The plant *Swertia mileensis* contains swerilactones (**42**), which inhibit HBV DNA replication, with IC_50_ values ranging from 1.53 to 5.34 μM/L [55]. Wang et al. extracted xanthone (**43**) and secoiridoid lactone (**44**) from *Swertia punicea*, and investigated their anti-HBV effects [56]. They found that xanthone and secoiridoid lactone inhibited HBV DNA replication in HepG2.2.15 cells, with IC_50_ values of 0.18 and 0.19 mM, respectively [56]. The plant *Herpetospermum caudigerum* contains herpetolide A [57]. Zhong et al. prepared a herpetolide A (**45**) nanosuspension lyophilized powder (HPA-NS-LP) and evaluated its anti-HBV effects in HepG2.2.15 cells [58]. They found that HPA-NS-LP significantly decreased HBV DNA levels [58]. Swertisin (**46**), isolated from the traditional Chinese medicinal herb *Iris tectorum* Maxim, significantly downregulated HBV DNA production in a dose-dependent manner in HepG2.2.15 cells, a de novo HepG2-NTCP cell infection model, and HBV transgenic mice [59].

### 2.6. Lignan Compounds

The non-flavonoid polyphenols known as lignans are widely distributed in the plant kingdom. Lignans exhibit antiviral, antioxidant, antibacterial, and antifungal activities in animal models. Lignans have diverse and complex chemical structures, although they are essentially dimers of phenylpropanoid units (C_6_-C_3_) linked by the central carbons of their side chains [60]. The chemical structures of some lignan compounds that affect HBV DNA replication are shown in Figure 6. The methanol extract of *Streblus asper* roots contains honokiol (**47**) and (7′*R*, 8′*S*, 7′*R*, 8′*S*)-erythron-strebluslignanol G (**48**), which reportedly show significant anti-HBV activity. These compounds significantly inhibit HBV DNA replication in Hep G2.2.15 cells transfected with HBV, with IC_50_ values of 9.02 and 8.67 μM, respectively [61,62]. Zhou et al., isolated cycloolivil-4-*O*-*β*-D-glucopyranoside (**49**) from *Stereospermum cylindricum* and evaluated its anti-HBV effects in HepG2.2.15 cells [63]. The compound was found to inhibit HBV DNA replication, with an IC_50_ value of 0.29 ± 0.034 mM (SI = 4.66) [63].

### 2.7. Xanthone Compounds

Xanthones are cyclic oxygenated compounds that display numerous bioactive properties, including antimicrobial, antitubercular, antitumor, antiviral, and antioxidant properties. Their health-promoting effects are mainly attributed to their tricyclic scaffold [64]. The chemical structures of some xanthone compounds affecting HBV DNA replication are shown in Figure 7. Xanthone compounds (**50**–**52**) extracted from *Curcuma xanthorrhiza* exhibit inhibitory effects at the post-entry stage of HBV infection. The effects of this plant extract were evaluated in Hep38.7 Tet cells; HBV DNA was quantified using an IC-RT-qPCR assay and the percentage inhibition was determined compared to that in the untreated control group. At 50 μg/mL, the *C. xanthorrhiza* extract reduced HBV DNA levels by 30% [65]. The xanthone compound mangiferin (**53**), isolated from *Swertia mussotii*, also induced a significant reduction in HBV DNA replication, with IC_50_ values ranging from 0.01 to 0.13 mM [66,67]. Cao et al. reported that dihydroxy-3,5-dimethoxyxanthone (**54**) and norswertianolin (**55**) isolated from *Swertia yunnanensis* exhibit inhibitory effects against HBV DNA replication in HepG2.2.15 cells, and methylation or glycosylation of the hydroxyl group of the compounds might be responsible for this inhibitory effect [38,66]. Another xanthone compound, 1,5,8-Trihydroxy-3-methoxyxanthone (**56**), isolated from *Swertia delavayi*, exhibited significant inhibitory activity against HBV DNA replication, with IC_50_ values of 0.09 and 0.05 mM/L (SI of 10.89) [66,68].

### 2.8. Tropolone Compounds

Tropolone is a seven-membered aromatic-ringed compound that includes a cyclic ketone functional group (cyclohepta-2,4,6-trien-1-one substituted by a hydroxy group at position 2) (**57**) (Figure 8). It is a toxin produced by the agricultural pathogen *Burkholderia plantarii*. It is an antibacterial and antifungal compound [69]. *β*-Thujaplicinol (**58**), extracted from Western Red Cedar heartwood (*Thuja plicata*, *Thuja occidentalis*, and *Chamaecyparis obtusa*), reportedly suppressed the replication of HBV strains of genotypes A and D in Huh7 cells transfected with HBV replication-competent plasmids by inhibiting RNAseH activity, with an estimated EC_50_ of 5 μM/L and a 50% cytotoxic concentration (CC_50_) of 10.1 mM/L [70].

### 2.9. Polysaccharide Compounds

Polysaccharides are natural polymeric compounds that exist as starch or cellulose in plants. Recent studies on polysaccharides have shown that they exhibit a wide range of biological activities, including immune enhancing, antiviral, and anti-inflammatory effects, as well as anti-HBV effects. Des(rhamnosyl) verbascoside (3,4-dihydroxybenzene ethanol-4-O-caffeoyl-β-D-glucoside) (**59**) is the main phenylethanol glycoside found in *Lindernia ruellioides* (Colsm.) Pennell (Figure 9), and its anti-HBV effects were evaluated in HepG2.2.15 cells. It significantly downregulated the expression of the HBV X protein (HBx) and inhibited DNA replication in a dose-dependent manner. In addition, it improved cell survival following H_2_O_2_-induced hepatocyte injury [71]. Another polysaccharide, heteropolysaccharide (FP-1), isolated from flaxseed hull using the hot water extraction method, reportedly exerted immunomodulatory effects by upregulating the mRNA expression levels of TNF-α, nitric oxide (NO), IL-6, and IL-12 in murine macrophages, and inhibited HBV DNA replication in HepG2.2.15 cells [72].

### 2.10. Others

Several compounds that could not be categorized under the classes described above have also been reported to decrease HBV DNA replication (Figure 10). Coumarin, which is distributed in various plant varieties, was isolated by Huang et al. from *Microsorum fortunei* (Moore) (**60**–**62**), and its anti-HBV effects were evaluated in HepG2.2.15 cells and DHBV-infected ducklings [73]. HBV quantification in the coumarin-treated cells and ducklings revealed a significant decrease in HBV DNA compared to the control groups [73]. *Cananga odorata* (**63**), an Indonesian plant, was shown to reduce HBV DNA and HBsAg secretion in Hep38.7-Tet cells, with an IC_50_ of 56.5 μg/mL [74].

## 3. Limitations of Previous Studies

The majority of studies that focused on screening phytomedicines and phytochemicals for downregulating HBV replication are mostly preliminary (Table 1 and Table 2). Several studies have demonstrated the anti-HBV effects of total plant extracts. However, the specific compounds responsible for these effects, as well as their structures, have not been determined. In addition, most of the studies were conducted using in vitro systems such as hepatoma cells transiently transfected with HBV replication-competent plasmids or stably HBV-producing HepG2.2.15 cells. Only a few of the studies were conducted using in vivo models such as Peking ducklings infected with DHBV or in transgenic mice. Another significant limitation of almost all these studies is that none of them elucidated the precise mechanism underlying the HBV DNA replication-inhibitory effects of the screened phytomedicines and phytochemicals. The HBV replication process is highly complex and has several steps (Figure 11). Some of the most important steps in HBV DNA replication and virion production include the production of cccDNA from partial dsDNA, the transcription of the cccDNA template leading to the four RNA strands, pgRNA encapsidation, reverse transcription within the capsid, synthesis of (+) strand DNA within the capsid, assembly of surface proteins, and virion release. Phytomedicines and phytochemicals that interfere with any of these steps of HBV DNA replication and virion production may decrease DNA levels and the virion copy number. However, almost none of the studies reviewed here investigated or described the specific step at which the compounds exerted their effects. The mechanisms by which these phytomedicines and phytochemicals affect HBV-derived factors and signaling pathways, as well as the host immune system, are yet to be investigated.

## 4. Future Approaches

Several studies have focused on evaluating the viral DNA replication-associated anti-HBV effects of phytomedicines and phytochemicals. However, accurate structures of the identified compounds, experimental animals with in vivo and de novo infection conditions, or clinical trials in humans are lacking. Moreover, the exact mechanisms of action of these compounds have not been elucidated. Thus, further studies should be performed to determine the specific structures of compounds that have been reported to exert significant inhibitory effects against HBV DNA replication. The specific steps of the HBV DNA replication process affected by the different compounds should be investigated by Southern blotting, Northern blotting, or Western blotting in combination with quantitative PCR. The stimulation of host factors or signaling pathways and immune system in the phytomedicines/phytochemicals treated and replicating HBV cell and host need to be properly investigated. In vivo investigations in experimentally HBV-infected animal models such as mice, monkeys, or chimpanzees, treated with different drugs of plant origin, would be the ideal approach for the clinical examination of the effects of these phytomedicines and phytochemicals. Finally, a comparative study using the most effective phytomedicines and phytochemicals in in vitro and in vivo systems is needed before validating their effects in HBV-infected patients.

## Figures and Tables

**Figure 1 ijms-23-01617-f001:**
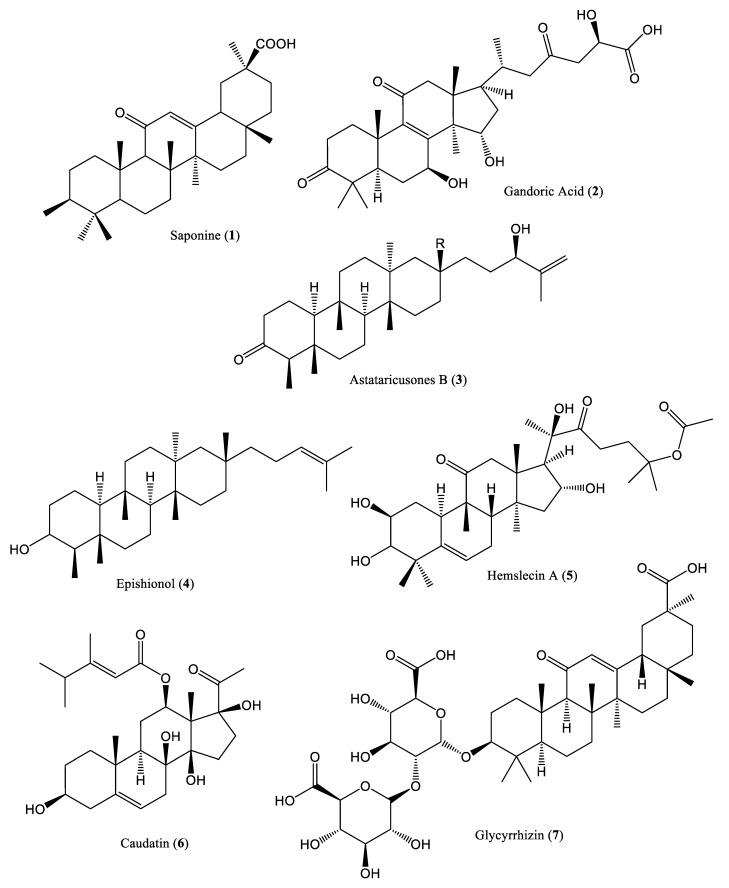
Chemical structures of different terpenoid compounds affecting HBV DNA replication.

**Figure 2 ijms-23-01617-f002:**
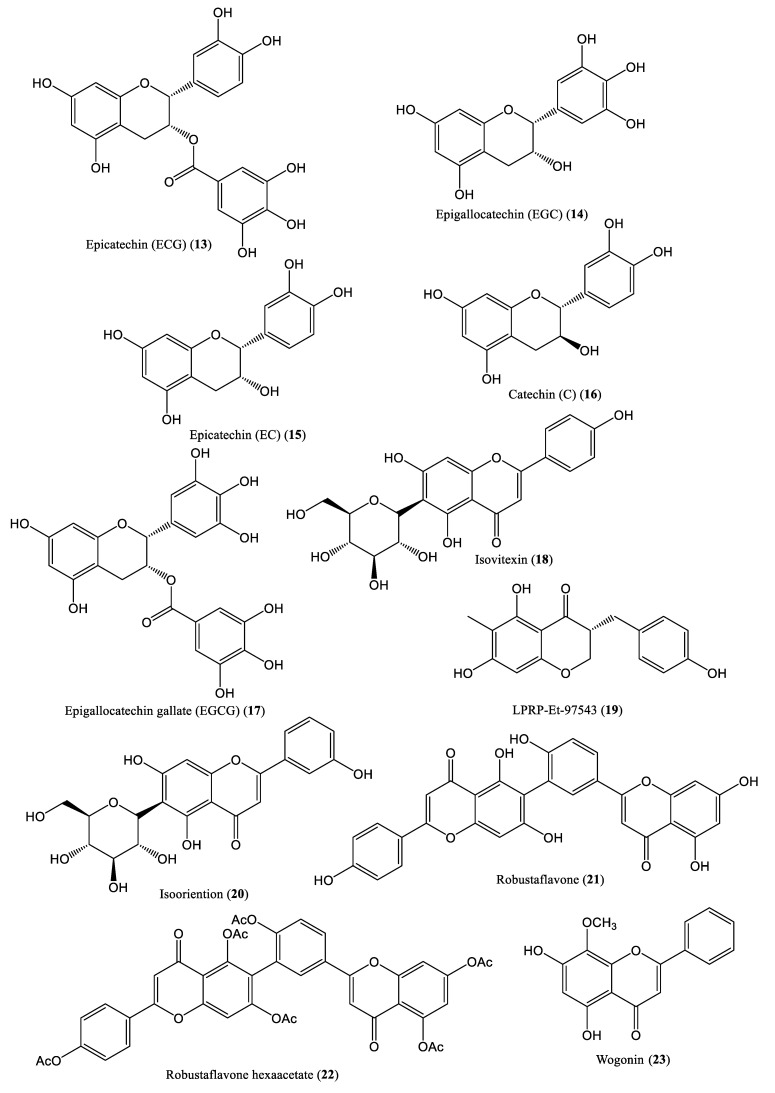
Chemical structures of flavonoid compounds affecting HBV DNA replication.

**Figure 3 ijms-23-01617-f003:**
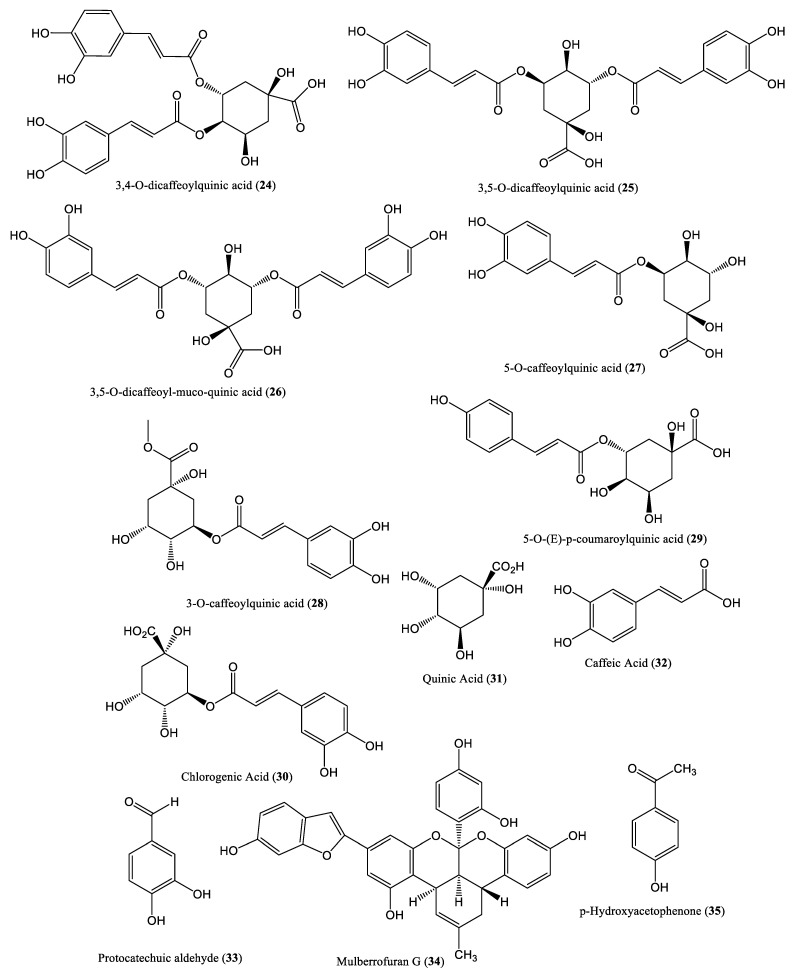
Chemical structures of some phenolic and polyphenolic compounds of plant origin that affect HBV DNA replication.

**Figure 4 ijms-23-01617-f004:**
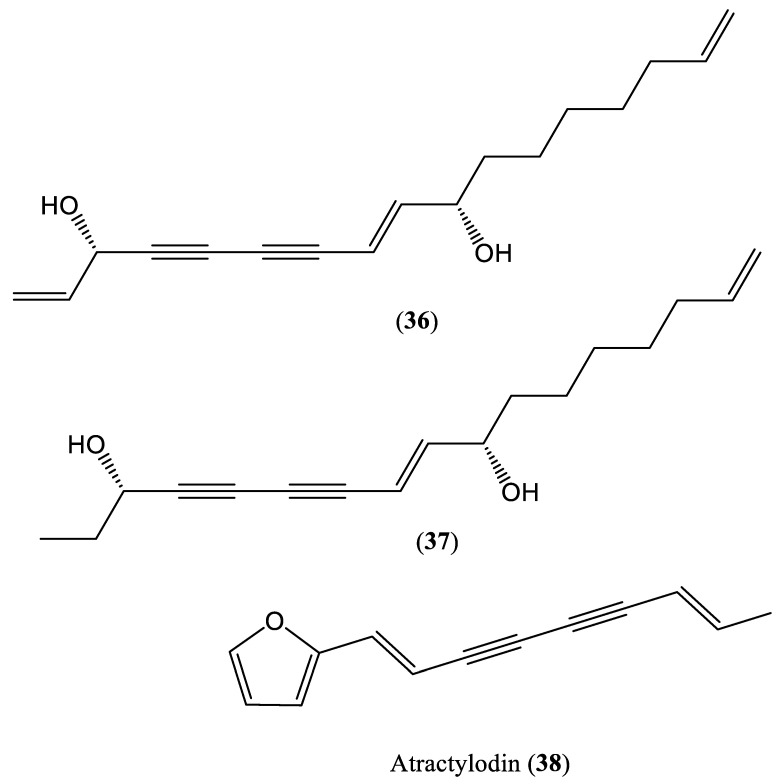
Chemical structures of some enyne compounds affecting HBV DNA replication.

**Figure 5 ijms-23-01617-f005:**
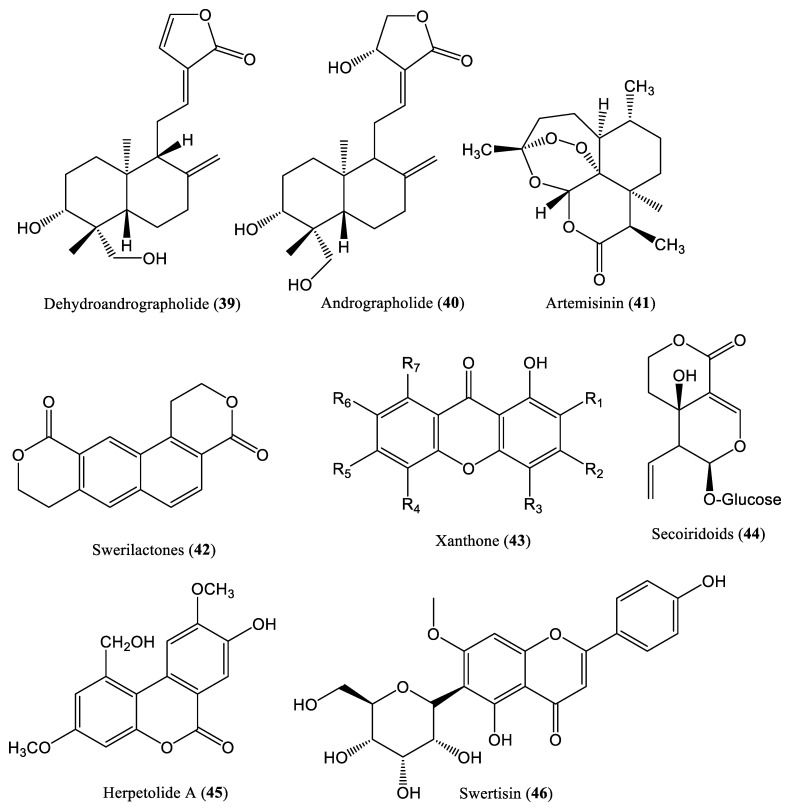
Chemical structures of some lactone compounds affecting HBV DNA replication.

**Figure 6 ijms-23-01617-f006:**
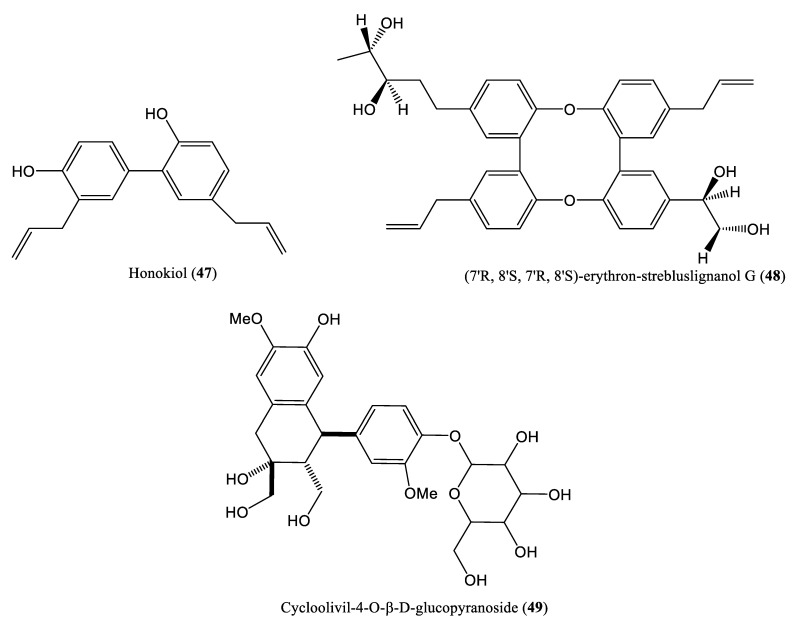
Chemical structures of some lignan compounds affecting HBV DNA replication.

**Figure 7 ijms-23-01617-f007:**
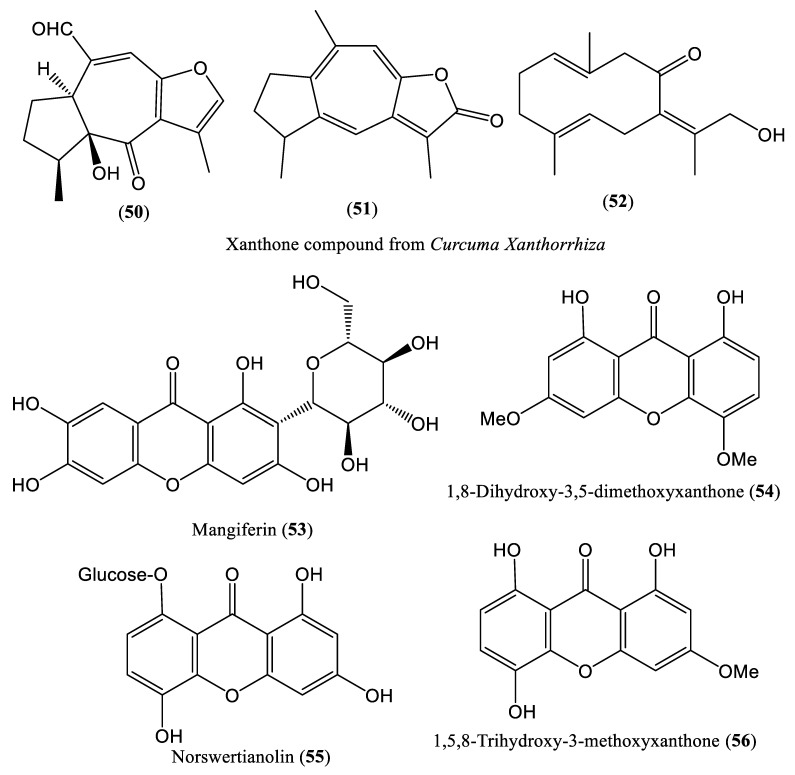
Chemical structures of some xanthone compounds affecting HBV DNA replication.

**Figure 8 ijms-23-01617-f008:**
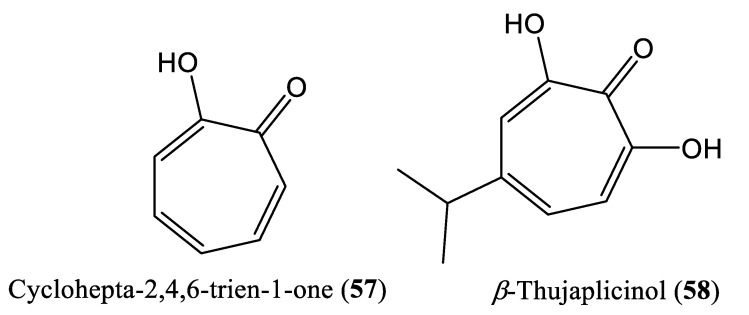
Chemical structures of some tropolone compounds affecting HBV DNA replication.

**Figure 9 ijms-23-01617-f009:**
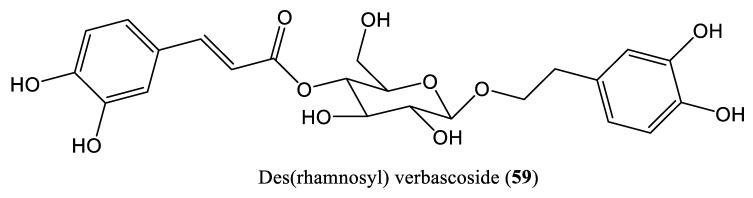
Chemical structure of a polysaccharide compound affecting HBV DNA replication.

**Figure 10 ijms-23-01617-f010:**
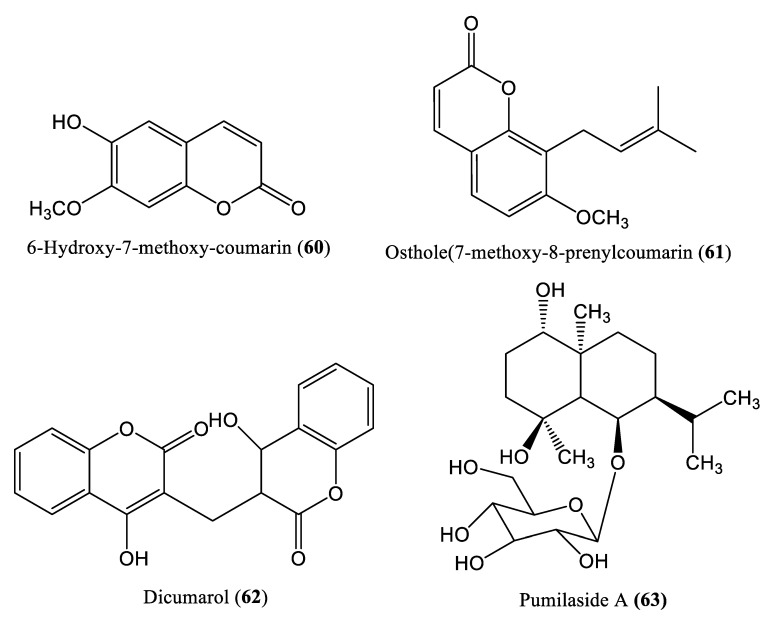
Chemical structures of other compounds affecting HBV DNA replication.

**Figure 11 ijms-23-01617-f011:**
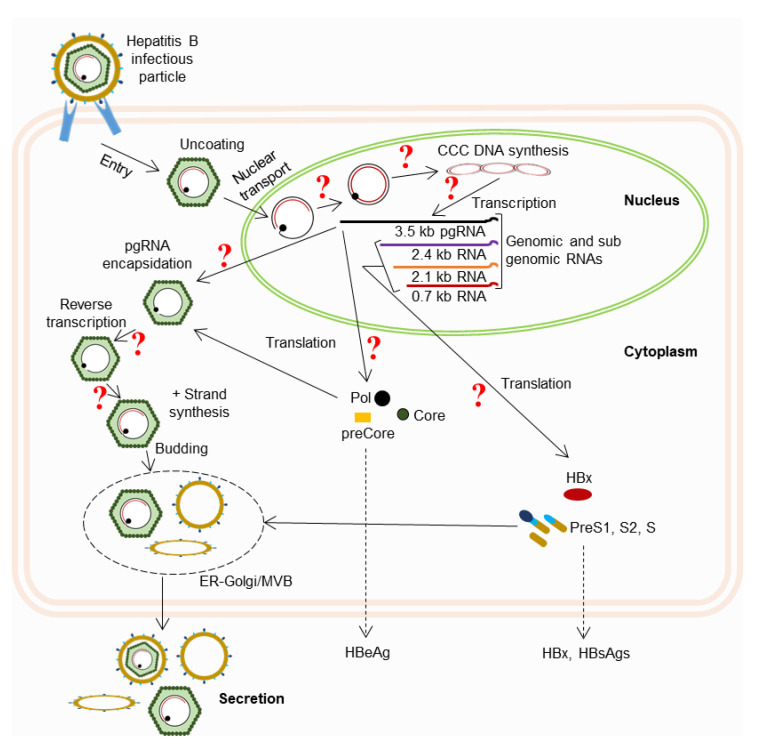
HBV replication cycle showing the possible steps in the DNA replication process that can be affected by phytomedicines/phytochemicals. The “?” mark indicates the possible steps where the phytomedicines may show inhibitory activity towards HBV DNA replication.

**Table 1 ijms-23-01617-t001:** Active compounds affecting HBV DNA replication and their sources.

Active Ingredient/ Compound	Concentrations and/or IC_50_	Mechanisms	Systems	Group	Plants/Sources	References
Saponin	40 μg/mL	Reduce DNA replication	HepG2.2.15 cells and Peking ducklings	Terpenoids	*Potentilla anserina*	[24]
Saponin	60 µg/mL	Inhibit DNA production	HepG2.2.15 cells and C57BL/6 mice	Terpenoids	*Abrus cantoniensis* Hance	[25]
Ganoderic acid	8 µg/mL	Inhibit HBV replication	HepG2.2.15 cells	Terpenoids	*Ganoderma lucidum*	[26]
Astataricusones B	2.7 μM	Inhibit DNA replication	HepG 2.2.15 cells	Terpenoids	*Aster tataricus*	[27]
Epishionol	30.7 μM	Inhibit DNA replication	HepG 2.2.15 cells	Terpenoids	*Aster tataricus*	[27]
Hemslecin A	11.2 μM	Inhibit DNA replication	HepG2.2.15 cells	Terpenoids	*Hemsleya*	[28]
Caudatin	40.62 mM	Inhibit DNA replication	HepG 2.2.15 cells	Terpenoids	*Cynanchum auriculatum*	[29]
Glycyrrhizin and glycyrrhetinic acid	39.28 μM/L	Inhibit DNA replication	HepG 2.2.15 cells	Terpenoids	*Glycyrrhizae glabra*	[30]
Helioxanthin	5.0 μM	Decrease DNA binding activity	HepA2 cells	Terpenoids	*Taiwania cryptomerioides*	[31]
Asiaticoside	150.0 μM	Reduce HBV transcription, replication, and cccDNA levels	HepG2.2.15 cells and ducklings	Terpenoids	*Hydrocotyle sibthorpioides*	[32]
Methyl helicterate	15.8 μM	Reduce cccDNA and RNA levels	HepG2.2.15 cells and ducklings	Terpenoids	*Helicteres angustifolia*	[33]
Betulinic acid	16.7 μM	Exhibit high binding affinities for the virus	Rat models and HepG2 cells	Terpenoids	*T. conophorum*	[34]
Epigallocatechin gallate (EGCG)	7.34 μM	Reduce cccDNA production	HepG2.117 cells	Flavonoids	*Camellia sinensis*	[36,37]
Isovitexin	0.09 mM	Suppresses DNA replication	HepG 2.2.15 cells	Flavonoids	*Swertia yunnanensis*	[38]
LPRP-Et-97543	10 μg/mL	Suppress core, S, and preS promoters and DNA replication	HepG2.2.15 cells and Huh7 cells	Flavonoids	*Liriope muscari*	[39]
Isooriention	0.02 mM	Inhibit DNA replication	HepG2.2.15 cells	Flavonoids	*Swertia mussotii*	[40]
Robustaflavone	0.25 mM	Inhibit DNA replication	HepG2.2.15 cells	Flavonoids	*Rhus succedanea*	[41]
Wogonin	0.73 mM	Reduce relaxed circular and linear forms of DNA	HBV-producing cell line (MS-G2)	Flavonoids	*Scutellaria baicalensis*	[42]
Polyphenolic extract	65.60 μg/mL	Reduced DNA level	HepG2.2.15 cells and ducklings	Phenolic and polyphenolic	*Geranium carolinianum* L.	[44]
Phenolic extract	0.40 g/L	Inhibit DNA level	HepG2.2.15 cells and ducks	Phenolic and polyphenolic	*Oenanthe javanica*	[45]
Quinic acid derivatives	-	Decrease DNA levels	HepG2.2.15 cells	Phenolic and polyphenolic	*Lactuca indica* L.	[46]
Chlorogenic acid	>1000 μM	Reduce single-stranded form of HBV DNA	HepG2.2.15 cells and ducklings	Phenolic and polyphenolic	Dicotyledonous plants	[47]
Protocatechuic aldehyde	100 mg/kg	Reduce DNA release	HepG2.2.15 cells and ducks	Phenolic and polyphenolic	*Salvia miltiorrhiza*	[48]
Mulberrofuran G	3.99 μM	Inhibit DNA replication	HepG2.2.15 cells	Phenolic and polyphenolic	*Morus alba* L.	[49]
*p*-Hydroxyacetophenone	306.4 μM	Inhibit DNA replication	HepG2.2.15 cells	Phenolic and polyphenolic	*Artemisia morrisonensis*	[50]
Atractylodin	9.8 μM	Inhibit DNA replication	HepG2.2.15 cells	Eneynes	*Artemisia capillaris*	[51]
Dehydroandrographolide and andrographolide	54.07 μM	Inhibit DNA replication	HepG2.2.15 cells	Lactones	*Andrographis paniculata*	[53]
Artemisinin	>100 mM	Inhibit rcDNA forms	HepG2.2.15 cells	Lactones	*Artemisia annua*	[54]
Swerilactones H-K	5.34 μM	Inhibit DNA replication	HepG2.2.15 cells	Lactones	*Swertia mileensis*	[55]
Xanthrone and secoiridoid lactone	0.19 mM	Inhibit DNA replication	HepG2.2.15 cells	Lactones	*Swertia punicea*	[56]
Herpetolide A	12.5 mg/kg	Decrease DNA levels	HepG2.2.15 cells	Lactones	*Herpetospermum caudigerum*	[58]
Swertisin	125 μM	Downregulate DNA production	HepG2.2.15 cells and HBV transgenic mice	Lactones	*Iris tectorum* Maxim	[59]
honokiol and (7′*R*, 8′*S*, 7′*R*, 8′*S*)-erythron-strebluslignanol G	8.67 μM	Inhibit DNA replication	HepG2.2.15 cells	Lignans	*Streblus asper*	[61,62]
Cycloolivil-4-*O*-*β*-D-glucopyranoside	0.29 mM	Inhibit DNA replication	HepG2.2.15 cells	Lignans	*Stereospermum cylindricum*	[63]
Extract	50 μg/mL	Reduce DNA levels	Hep38.7 Tet cells	Xanthones	*Curcuma xanthorrhiza*	[65]
Mangiferin	0.13 mM	Inhibit DNA replication	HepG2.2.15 cells	Xanthones	*Swertia mussotii*	[66,67]
Dihydroxy-3,5-dimethoxyxanthone and norswertianolin	0.21 mM	Inhibit DNA replication	HepG2.2.15 cells	Xanthones	*Swertia yunnanensis*	[38,66]
Trihydroxy-3-methoxyxanthone	0.05 mM	Inhibit DNA replication	HepG2.2.15 cells	Xanthones	*Swertia delavayi*	[66,68]
*β*-Thujaplicinol	10.1 mM	Inhibit RNAseH activity	Huh7 cells	Tropolone	*Thuja plicata*, *Thuja occidentalis* and *Chamaecyparis obtusa*	[70]
Des(rhamnosyl) verbascoside	12.5 mg/L	Downregulate HBx and inhibit DNA replication	HepG2.2.15 cells	Polysaccharide	*Lindernia ruellioides*	[71]
heteropolysaccharide (FP-1)	250 μg/mL	Inhibit DNA replication	HepG2.2.15 cells	Polysaccharide	Flaxseed hull	[72]

**Table 2 ijms-23-01617-t002:** Approaches and limitations of the investigations on phytomedicines affecting HBV DNA replication.

Studies	Approaches (Compound and Methods)	Limitations	References
Zhao et al., 2008	Saponin HepG2.2.15 cells and Peking ducklings	No investigation on specific step of HBV DNA replication No mammalian infection (in vivo) system is used	[24]
Yao et al., 2020	Saponin HepG2.2.15 cells and C57BL/6 mice	No investigation on specific step of HBV DNA replication Only HBV genome containing recombinant adenovirus is used in mammalian infection system	[25]
Li and Wang, 2006	Ganoderic acid HepG2.2.15 cells	No investigation on specific step of HBV DNA replication	[26]
Zhou et al., 2013	Astataricusones B and epishionol HepG2.2.15 cells	Only cell culture system (in vitro) is used No investigation on specific step of HBV DNA replication No mammalian infection (in vivo) system is used	[27]
Guo et al., 2013	Hemslecin A HepG2.2.15 cells	Only cell culture system (in vitro) is used No investigation on specific step of HBV DNA replication No mammalian infection (in vivo) system is used	[28]
Wang et al., 2012a	Caudatin HepG2.2.15 cells	Only cell culture system (in vitro) is used No investigation on specific step of HBV DNA replication No mammalian infection (in vivo) system is used	[29]
Wang et al., 2012b	Glycyrrhizin and glycyrrhetinic acid HepG2.2.15 cells	Only cell culture system (in vitro) is used No investigation on specific step of HBV DNA replication No mammalian infection (in vivo) system is used	[30]
Tseng et al., 2008	Helioxanthin HepA2 cells	Only cell culture system (in vitro) is used No mammalian infection (in vivo) system is used	[31]
Huang et al., 2013b	Asiaticoside HepG2.2.15 cells and ducklings	No mammalian infection (in vivo) system is used	[32]
Huang et al., 2013a	Methyl helicterate HepG2.2.15 cells and ducklings	No mammalian infection (in vivo) system is used	[33]
Song, 2018; Ye et al., 2009	Epigallocatechin gallate (EGCG) HepG2.117 cells	Only cell culture system (in vitro) is used No mammalian infection (in vivo) system is used	[36,37]
Cao et al., 2013a	Isovitexin HepG2.117 cells	Only cell culture system (in vitro) is used No investigation on specific step of HBV DNA replication No mammalian infection (in vivo) system is used	[38]
Huang et al., 2014	LPRP-Et-97543 HepG2.2.15 cells and Huh7 cells	Only cell culture system (in vitro) is used No mammalian infection (in vivo) system is used	[39]
Cao et al., 2015b	Isooriention HepG2.2.15 cells	Only cell culture system (in vitro) is used No investigation on specific step of HBV DNA replication No mammalian infection (in vivo) system is used	[40]
Zembower et al., 1998	Robustaflavone HepG2.2.15 cells	Only cell culture system (in vitro) is used No investigation on specific step of HBV DNA replication No mammalian infection (in vivo) system is used	[41]
Huang et al., 2000	Wogonin HBV-producing cell line (MS-G2)	Only cell culture system (in vitro) is used No mammalian infection (in vivo) system is used	[42]
Li et al., 2008	Polyphenolic extract HepG2.2.15 cells and ducklings	No investigation on specific step of HBV DNA replication No mammalian infection (in vivo) system is used	[44]
Han et al., 2008	Phenolics extract HepG2.2.15 cells and ducks	No investigation on specific step of HBV DNA replication No mammalian infection (in vivo) system is used	[45]
Kim et al., 2007	Quinic acid derivatives HepG2.2.15 cells	Only cell culture system (in vitro) is used No investigation on specific step of HBV DNA replication No mammalian infection (in vivo) system is used	[46]
Wang et al., 2009	Chlorogenic acid HepG2.2.15 cells and ducklings	No mammalian infection (in vivo) system is used	[47]
Zhou et al., 2007	Protocatechuic aldehyde HepG2.2.15 cells and ducks	No investigation on specific step of HBV DNA replication No mammalian infection (in vivo) system is used	[48]
Geng et al., 2012	Mulberrofuran G HepG2.2.15 cells	Only cell culture system (in vitro) is used No investigation on specific step of HBV DNA replication No mammalian infection (in vivo) system is used	[49]
Zhao et al., 2015	*p*-Hydroxyacetophenone HepG2.2.15 cells	Only cell culture system (in vitro) is used No investigation on specific step of HBV DNA replication No mammalian infection (in vivo) system is used	[50]
Geng et al., 2018	Atractylodin HepG2.2.15 cells	Only cell culture system (in vitro) is used No investigation on specific step of HBV DNA replication No mammalian infection (in vivo) system is used	[51]
Chen et al., 2014	Dehydroandrographolide and andrographolide HepG2.2.15 cells	Only cell culture system (in vitro) is used No investigation on specific step of HBV DNA replication No mammalian infection (in vivo) system is used	[53]
Romero et al., 2005	Artemisinin HepG2.2.15 cells	Only cell culture system (in vitro) is used No mammalian infection (in vivo) system is used	[54]
Geng et al., 2011	Swerilactones H-K HepG2.2.15 cells	Only cell culture system (in vitro) is used No investigation on specific step of HBV DNA replication No mammalian infection (in vivo) system is used	[55]
Wang et al., 2013	Xanthrone and secoiridoid lactone HepG2.2.15 cells	Only cell culture system (in vitro) is used No investigation on specific step of HBV DNA replication No mammalian infection (in vivo) system is used	[56]
Zhong et al., 2020	Herpetolide A HepG2.2.15 cells	Only cell culture system (in vitro) is used No investigation on specific step of HBV DNA replication No mammalian infection (in vivo) system is used	[58]
Xu et al., 2020	Swertisin HepG2.2.15 cells and HBV transgenic mice	No investigation on specific step of HBV DNA replication HBV producing transgenic mice is used No *de novo* infection in mice is performed	[59]
Li et al., 2012; Li et al., 2013	Honokiol and (7′*R*, 8′*S*, 7′*R*, 8′*S*)-erythron-strebluslignanol G HepG2.2.15 cells	Only cell culture system (in vitro) is used No investigation on specific step of HBV DNA replication No mammalian infection (in vivo) system is used	[61,62]
Zhou et al., 2015	Cycloolivil-4-*O*-*β*-D-glucopyranoside HepG2.2.15 cells	Only cell culture system (in vitro) is used No investigation on specific step of HBV DNA replication No mammalian infection (in vivo) system is used	[63]
Tutik et al., 2020	Extract Hep38.7 Tet cells	Only cell culture system (in vitro) is used No investigation on specific step of HBV DNA replication No mammalian infection (in vivo) system is used	[65]
Cao et al., 2013b; Liu et al., 2020	Mangiferin HepG2.2.15 cells	Only cell culture system (in vitro) is used No investigation on specific step of HBV DNA replication No mammalian infection (in vivo) system is used	[66,67]
Cao et al., 2013a; Liu et al., 2020	Dihydroxy-3,5-dimethoxyxanthone and norswertianolin HepG2.2.15 cells	Only cell culture system (in vitro) is used No investigation on specific step of HBV DNA replication No mammalian infection (in vivo) system is used	[38,66]
Cao et al., 2015a; Liu et al., 2020	Trihydroxy-3-methoxyxanthone HepG2.2.15 cells	Only cell culture system (in vitro) is used No investigation on specific step of HBV DNA replication No mammalian infection (in vivo) system is used	[66,68]
Hu et al., 2013	*β*-Thujaplicinol Huh7 cells	Only cell culture system (in vitro) is used No mammalian infection (in vivo) system is used	[70]
Mou et al., 2021	Des(rhamnosyl) verbascoside HepG2.2.15 cells	Only cell culture system (in vitro) is used No mammalian infection (in vivo) system is used	[71]
Liang et al., 2019	Heteropolysaccharide (FP-1) HepG2.2.15 cells	Only cell culture system (in vitro) is used No investigation on specific step of HBV DNA replication No mammalian infection (in vivo) system is used	[72]

## Data Availability

Not applicable.

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
