# Peer review of "Phytomedicines to Target Hepatitis B Virus DNA Replication: Current Limitations and Future Approaches"

_ijms, 2022, doi:10.3390/ijms23031617_

Round 1
Reviewer 1 Report
Authors do not comment on the mechanism of HBV inhibition by phytomedicine related compounds. In Figure1 they just use "?" the effect in HBV life cycles. Moreover, this review manuscript just introduces limitations of assessing HBV inhibition mechanism nevertheless they showed structure in each compound.
Author Response
Thank you very much for reviewing our manuscript and for your comments and criticisms.
The HBV replication process is highly complex and has several steps (Figure 1). Some of the most important steps in HBV DNA replication and virion production include the production of cccDNA from partial dsDNA, the transcription of cccDNA template leading to the four RNA strands, pgRNA encapsidation, reverse transcription within the capsid, synthesis of (+) strand DNA within the capsid, assembly of surface proteins, and virion release. Phytomedicines and phytochemicals that interfere with any of these steps of the HBV DNA replication and virion production may decrease DNA levels and virion copy number. However, almost none of the studies reviewed here investigated or described the specific step at which the compounds exerted their effects. The mechanisms by which these phytomedicines and phytochemicals affect HBV-derived factors and signaling pathways, as well as the host immune system, are yet to be investigated. We also discussed this in the manuscript (Page 18; Lines: 341-352).
Therefore, we tried to show the possible steps in the HBV DNA replication process that can be affected by phytomedicines/phytochemicals. The “?” mark indicates the possible steps where the phytomedicines may show the inhibiting capacity to HBV DNA replication which need to be investigated in the future.
We also included table 2 (As suggested by the Academic Editor) showing the limitations of the previous investigations

Reviewer 2 Report
It is an interesting article that illustrates the research of scientists for alternative medicines against drug-resistant HBV.
Many future studies are needed to demonstrate the effectiveness of phytomedicines and phytochemicals against HBV in real life.
Specify the meaning of the abbreviations IC50 (row129 and others).
Author Response
Thank you very much for reviewing our manuscript and for your appreciation and comments.
The meaning of the abbreviations IC50 is “half-maximal inhibitory concentration”. We now mentioned in the revised manuscript (Page: 4, Line: 129) and later the “IC50 “ is used as an abbreviation.

Reviewer 3 Report
This manuscript provides an extensive review of the major groups of phyto-medicines and phyto-chemicals that present with the ability to interfer with the replication of HVB primarily in in-vitro cell culture models of liver cells (HepG2 cells but also HepA2 or Hep2.2.15) transfected with HBV, and in some in-vivo models (ducklings). The results reviewed in the manuscript appear to be promising but the limited amount of data in in-vivo models labels the result as mostly preliminary.
Overall, the review article is properly written and addresses the main families of plant extracts that present protective effects.
No major criticisms were noted. THe limitations of the studies reported in the review are correctly identified by the authors.
Author Response
Thank you very much for reviewing our manuscript and for your appreciation and comments.
We agreed with you that the limited amount of data in in vivo models labels the result as mostly preliminary. This is one of the most important limitations of the previous studies. Therefore, our current review article will be very helpful for future in-depth investigations for de novo infection in in vivo conditions.

Round 2
Reviewer 1 Report
Authors corrected in this revised manuscript and newly added Table2 about the limitation of phytomedicines for HBV infected cells. These explanations of limitations are just criticizing previous publication and readers cannot incorporate any information regarding HBV antiviral effect. Moreover, Figure1 does not have any helpful information for the effect of phytomedicines. Finally, this manuscript has few information as review article.
Author Response
Thank you very much for reviewing our manuscript and for your comments and criticisms.
We believe that the readers can easily understand the limitations of the previous studies on phytomedicines for HBV treatment from the Table 2 and the scientists can use this information for future planning and investigation.
Figure 1 demonstrated the possible steps in the HBV DNA replication process that can be affected by phytomedicines/phytochemicals. The “?” mark indicates the possible steps where the phytomedicines may show the inhibiting capacity to HBV DNA replication which need to be investigated in the future.
This manuscript is a resubmission of an earlier submission. The following is a list of the peer review reports and author responses from that submission.
Round 1
Reviewer 1 Report
Authors introduced various phytomedicines that exhibit the experimental effect for reduction of HBV-DNA in the host and noted that these drugs express effectiveness for NA resistant HBV and little adverse effects. Moreover, some of phytomedicines might reduce HBV cccDNA as well as HBV-DNA. Though this review looks like very informative since there are few review reports regarding phytomedicines so far, the effect and antiviral mechanism are varied, and, confusing. Authors need to introduce so that these compounds are promising medicines for HBV.
Major points.
- Authors introduced in the Table 1 phytomedicine related compounds, but not compared HBV-DNA and HBV cccDNA reduction in each drugs just listed Ref. No.. They need to include more information about pharmacological effects such as IC50, the degree of HBV-DNA reduction whether there is HBV cccDNA reduction in each compound, and the mechanism in the HBV life cycles as shown in Figure1.
- Authors showed the chemical structure of each compound in several figure; however, it needs to show what structure contributes HBV reduction.
Author Response
Authors introduced various phytomedicines that exhibit the experimental effect for reduction of HBV-DNA in the host and noted that these drugs express effectiveness for NA resistant HBV and little adverse effects. Moreover, some of phytomedicines might reduce HBV cccDNA as well as HBV-DNA. Though this review looks like very informative since there are few review reports regarding phytomedicines so far, the effect and antiviral mechanism are varied, and, confusing. Authors need to introduce so that these compounds are promising medicines for HBV
Thank you very much for reviewing our manuscript and for your appreciation and comments. Your excellent suggestions and comments helped us to improve the manuscript quality.
Major points
Point 1: Authors introduced in the Table 1 phytomedicine related compounds, but not compared HBV-DNA and HBV cccDNA reduction in each drugs just listed Ref. No.. They need to include more information about pharmacological effects such as IC50, the degree of HBV-DNA reduction whether there is HBV cccDNA reduction in each compound, and the mechanism in the HBV life cycles as shown in Figure 1.
Response 1: Mechanisms and systems used for the determination of the anti-HBV effect of various phytomedicines were included in the table (table 1). However, pharmacological effects such as IC50 are greatly varied in each drug and experiment. In addition, several concentrations of each phytomedicine were tested and accordingly degrees of reduction of HBV DNA level was varied. On the other hand, IC50 had not been used always, some investigations calculated as mg/L, μg/ML. The DNA reduction was also calculated differently in different studies such as percentage of qPCR data, Southern blot, Northern blot, etc. It would be very complex if the several concentrations of each phytomedicine and their effects on HBV replication are included in the table. However, we described these points of each phytomedicine in the text (Please see the red font text in the manuscript). We hope you will understand the points.
Point 2: Authors showed the chemical structure of each compound in several figure; however, it needs to show what structure contributes HBV reduction.
Response 2: Thank you very much for your nice suggestion. It would be excellent if we could do it. But unfortunately, most of the studies did not investigate which structure is responsible for the anti-HBV effect. We independently have drawn the chemical structures of the phytomedicines based on their group. The phytomedicines need to be synthesized based on the drawn structures and should be verified their effect in the laboratory by both in vitro and in vivo systems. Therefore, our review manuscript would be of great interest for further investigations.

Reviewer 2 Report
The aim of this manuscript is to review the most recent evidence on the significant inhibitory effects of phytomedicines/phytochemicals on HBV Virus DNA replication.
Even if the manuscript provides an organic overview, with a densely organized structure and based on well-synthetized data, there are aspects to be mentioned, to make the article fully readable. For these reasons, the manuscript requires minor changes.
Please find below an enumerated list of comments on my review of the manuscript:
INTODUCTION:
LINE 73: Hepatitis B Virus infection is a global health problem and represents a major cause of acute and chronic liver disease. To this aim, several and recent studies evaluate the persistence of a long-term immunogenicity of HBV vaccine, in order to established the main associated factors and determinants of the disease (see, for reference: Mastrodomenico, M.; Muselli, M.; Provvidenti, L.; Scatigna, M.; Bianchi, S.; Fabiani, L. Long-term immune protection against HBV: Associated factors and determinants. Hum. Vaccines Immunother. 2021).
LINE 78: Recent studies focuses on the vertical and horizontal transmission of HBV Virus, providing an epidemiological profile of HBV and also evaluating the long – term effects of the available vaccination (see, for reference: Chang MS, Nguyen MH. Epidemiology of hepatitis B and the role of vaccination. Best Pract Res Clin Gastroenterol. 2017 Jun;31(3):239-247. doi: 10.1016/j.bpg.2017.05.008. Epub 2017 Jun 6. PMID: 28774405).
Besides, the methodology design was rigorous and appropriately implemented within the study.
In conclusion, this manuscript is densely presented and well organized, based on well-synthetized data. The authors were lucid in their style of writing, making it easy to read and understand the message, portrayed in the manuscript. Moreover, the methodology design was rigorous and appropriately implemented within the study. However, many of the topics are very concisely covered. At the same time, this research have futuristic importance and could be potential for future research. However, I have minor comments only for the introductive section, for improvement before acceptance for publication. I would accept the manuscript, if the comments are addressed properly.
Author Response
The aim of this manuscript is to review the most recent evidence on the significant inhibitory effects of phytomedicines/phytochemicals on HBV Virus DNA replication.
Even if the manuscript provides an organic overview, with a densely organized structure and based on well-synthetized data, there are aspects to be mentioned, to make the article fully readable. For these reasons, the manuscript requires minor changes.
Please find below an enumerated list of comments on my review of the manuscript:
Thank you very much for reviewing our manuscript and for the appreciation and comments.
Point 1: LINE 73: Hepatitis B Virus infection is a global health problem and represents a major cause of acute and chronic liver disease. To this aim, several and recent studies evaluate the persistence of a long-term immunogenicity of HBV vaccine, in order to established the main associated factors and determinants of the disease (see, for reference: Mastrodomenico, M.; Muselli, M.; Provvidenti, L.; Scatigna, M.; Bianchi, S.; Fabiani, L. Long-term immune protection against HBV: Associated factors and determinants. Hum. Vaccines Immunother. 2021).
Response 1: The information has been included in the revised manuscript and the reference has been cited (Page: 2, lines: 71-74).
Point 2: LINE 78: Recent studies focuses on the vertical and horizontal transmission of HBV Virus, providing an epidemiological profile of HBV and also evaluating the long – term effects of the available vaccination (see, for reference: Chang MS, Nguyen MH. Epidemiology of hepatitis B and the role of vaccination. Best Pract Res Clin Gastroenterol. 2017 Jun;31(3):239-247. doi: 10.1016/j.bpg.2017.05.008. Epub 2017 Jun 6. PMID: 28774405).
Response 2: The information has been added in the revised manuscript and the reference has been cited (Page: 2, lines: 83-85).
Point 3: Besides, the methodology design was rigorous and appropriately implemented within the study.
Response 3: Thank you very much for your appreciation.
Point 4: In conclusion, this manuscript is densely presented and well organized, based on well-synthetized data. The authors were lucid in their style of writing, making it easy to read and understand the message, portrayed in the manuscript. Moreover, the methodology design was rigorous and appropriately implemented within the study. However, many of the topics are very concisely covered. At the same time, this research have futuristic importance and could be potential for future research. However, I have minor comments only for the introductive section, for improvement before acceptance for publication. I would accept the manuscript, if the comments are addressed properly.
Response 4: Thank you very much for your appreciation.

Round 2
Reviewer 1 Report
Comment 1; Authors added HBV related information in Table 1 and added IC50 information in each compound in the manuscript. Each IC50 is more informative if included in the table. Looking at these materials in each experiment, there is no report that used HBV infected primary hepatocyte, these results lead to be doubtful whether these HBV reductions are truly working as anti HBV activity, moreover authors do not promptly explain which step is important in each HBV viral cycle.
Comment 2; Authors do not discuss the HBV reduction mechanism from various phytomedicine related studies. It cannot be understood the priority the importance of the mechanism regarding HBV antiviral activity and readers cannot understand whether phytomedicine is useful or not.
Since authors do not discuss these commented matters, this manuscript is not meaningful for introducing HBV antiviral effect by phytomedicine related compound.